# Impact of Technique and Schedule of Reirradiation Plus Hyperthermia on Outcome after Surgery for Patients with Recurrent Breast Cancer

**DOI:** 10.3390/cancers11060782

**Published:** 2019-06-05

**Authors:** Sabine Oldenborg, Rob van Os, Bing Oei, Philip Poortmans

**Affiliations:** 1Department of Radiation Oncology, Amsterdam UMC, University of Amsterdam (AMC), 1105 AZ Amsterdam, The Netherlands; r.vanos@amsterdamumc.nl; 2Department of Radiation Oncology, Institute Verbeeten (BVI), 5042 SB Tilburg, The Netherlands; sboei@ziggo.nl; 3Department of Radiation Oncology, Institut Curie, Paris Sciences & Lettres-PSL University, 75248 Paris, France

**Keywords:** recurrent breast cancer, reirradiation, surgery, hyperthermia, clinical outcome, toxicity, fraction dose

## Abstract

Purpose: Combining reirradiation (reRT) with hyperthermia (HT) has shown to be of high therapeutic value for patients with loco-regionally recurrent breast cancer. The purpose of this study was to compare the long-term therapeutic effect and toxicity of reRT + HT following surgery of loco-regionally recurrent breast cancer using two different reRT regimens. Methods: The reRT regimen of the 78 patients treated in Institute A consisted of 8 × 4 Gy twice a week using mostly abutted photon-electron fields. The 78 patients treated in Institute B received a reRT regimen of 12 × 3 Gy, four times a week with single or multiple electron fields. Superficial hyperthermia was applied once a week in Institute A and twice a week in Institute B. Both institutes started HT treatment within 1 hour after reRT and used the same 434-MHz systems to heat the tumor area to 41–43 °C. Results: The 5-year-infield local control (LC) rates were similar; however, the 5-year-survival rates were 13% lower in Institute A. Most remarkable was the difference in risk with respect to 5-year ≥ grade 3 toxicity, which was more than twice as high in Institute A. Conclusion: The combination of reirradiation and hyperthermia after macroscopically complete excision of loco-regional breast cancer recurrences provides durable local control in patients at risk for locoregional recurrent breast cancer. Treatment is well tolerated with the 12 × 3 Gy schedule with limited-sized electron fields.

## 1. Introduction

Locoregional recurrence after mastectomy or breast conservation is associated with a poor outcome in patients with breast cancer [1,2,3]. Hyperthermia, the elevation of tumor temperature to 40–45 °C, is a well-established radio- and chemotherapy sensitizer [4,5,6,7]. Several phase III trials demonstrated a significant increase of complete response rates and duration of local control when hyperthermia (HT) was added to reirradiation (reRT) for locoregionally recurrent breast cancer in previously irradiated areas, without an significant increase in toxicity [8,9,10]. The Dutch National guidelines therefore adopted the combination of reRT + HT as standard of care for recurrent breast cancer in a previously irradiated area, preferably preceded by surgery [11]. Prospective trials to investigate the effect of reRT + HT in the “adjuvant setting” have never been performed. The information thereof remains restricted to data from only a few retrospective, often small-sized studies [12,13,14,15,16,17]. This information is too limited to determine the clinical value of using reRT + HT in this setting.

Back in the early days of hyperthermia in around the year 1980, RT and HT techniques were less sophisticated, with knowledge as well as experience far more limited. Then, even more than today, patients who had no other treatment options left had to be treated. However, they were at higher risk of developing severe side effects following reRT + HT as a cumulative result of the reRT + HT after previous treatments, probably also combined with individual sensitivity to toxicity. Prospective clinical studies on hyperthermia and reRT were and are difficult, due to the small number of hyperthermia institutes in the world and the limited availability of commercial equipment. In addition, it would be quite impossible to obtain sufficient patient accrual to reliably address the effects of RT technique and schedule on clinical outcome and toxicity.

The current study aims to, retrospectively, gain more insight in the influence of reRT technique and schedule amongst other factors, on long-term clinical outcome for patients treated with reRT + HT after macroscopically complete resection. Data from two different Dutch hyperthermia centers, Institute A and Institute B, were compared. We focused on severe late toxicity as one of the main endpoints because of its major impact on patient quality of life in comparison to mild toxicity. We have put our emphasis on the burden of the treatment with respect to the beneficial effects. Multivariable statistical analyses were carried out from to gain insight in factors influencing subsequent infield local control (LC) and late morbidity in order to contribute to future treatment improvements.

## 2. Methods and Materials

### 2.1. Patients

Results concerning 78 patients with locoregional recurrent breast cancer in areas previously irradiated with curative intent and treated between 1988 and 2001 in Institute A were published earlier by the first author (S. Oldenborg) [18]. Data have now been collected from patients treated between 1988 and 2005 with reRT and HT at Institute B. According to the Dutch National guidelines, all patients with locoregional breast cancer recurrences in previously irradiated areas should be treated with reirradiation + hyperthermia, preferably preceded by surgery. The inclusion criteria of both institutes, consequently, are the same. We included all these patients retrospectively, without exclusion. Patients received ReRT + HT during those periods as an adjunct to surgery based on the presence of assumed individual additional risk factors: (1) surgery results (microscopic residual tumor or close resection margin), (2) recurrence histology (tumor size, multi-centricity, diffuse tumor growth, angioinvasion or poor differentiation grade), (3) medical history and disease status (e.g., young age (<49 years), in primary treatment or with multiple previous recurrences at the same location). By coincidence, similar to Institute A, a total number of 78 consecutive patients were identified. All but one of the patients were female.

Data were collected from the radiation therapy and hyperthermia patient charts. In case of missing follow-up data, questionnaires were sent to referring specialists, and/or general practitioners. 

All patients previously received high-dose radiation, overlapping with the current reRT field. For the current recurrence episode all patients had a macroscopically complete surgery.

Characteristics of the current disease and treatment episode are summarized in Table 1. We only reported disease characteristics known for at least 80% of the population to be able to explore correlations with the endpoints. Unfortunately, despite our repetitive follow-up requests to referring specialist, only a few details about the performed surgery were recovered. This is partly due to the fact that a reasonable number of patients were treated at a time when some of today’s pathology tests like Human Epidermal frowth factor Receptor 2 (HER2-neu) receptor status were not yet routinely performed. Also, almost 50% of our patients had several surgical interventions either for their second, third, fourth, or even fifth recurrence; hence, pathological analysis was only used to confirm presence of disease or not.

### 2.2. Treatment

#### Radiation Therapy

Different treatment approaches were used in each center. In Institute A, patients were irradiated using a schedule of eight fractions of 4 Gy given twice a week to a total dose of 32 Gy [18]. In Institute B, the reRT schedule consisted of 12 fractions of 3 Gy given four times a week to a total dose of 36 Gy. For large target areas, Institute A used abutted photon and electron fields (regimen A; Figure 1a, adapted from our previous publication [19]). In general, the chest wall or mastectomy area up to the dorsal axillary fold was considered the target area, thereby including an elective volume because of the wide margins. A bolus was applied to reach the most superficial layers of the skin. Thickness was determined by radiation technique and energy and adjusted according to tumor depth for each patient individually. When single fields were applied the target volume was limited to the recurrence area with a 3–5 cm margin.

In Institute B, the clinical target volume included 2 to 2.5 cm around the resection area or around the recurrent tumor(s), leaving the radiation field borders at least 3 cm around scars/recurrences. In the case of large target volumes, a combination of 2–3 alternating abutted anterior-posterior (AP) electron fields were used. Fields were separated by gaps, creating a number of different fields. Alternatingly, either two or three fields were irradiated at once (regimen B). The distance between the alternating gap locations was adjusted for every individual patient to minimize both underdosage and overdosage, thereby obtaining a more homogenous dose distribution (Figure 1b) [19].

### 2.3. Hyperthermia

Again, different treatment approaches were used in each center. HT was given once a week in Institute A and twice a week in Institute B. In Institute A HT was started within 1 hour after reRT [18], in Institute B within 1 hour after reRT. In both centers heat was induced electromagnetically, using externally applied contact flexible microstrip applicators (STOK, Fryazino, Russia), operating at 434 MHz [20,21]. Treatment fields covered the entire target area. For all patients temperatures were measured with thin flexible multisensory thermocouple probes (ELLA CS, Hradec Králové, Czech Republic) on the skin and, if feasible or preferable, invasively using a flexible subcutaneous catheter. Aim-temperature was 41–43 °C for one hour. Applied power was adjusted to the desired temperature distribution without exceeding the maximum normal tissue temperatures (45 °C) or patient tolerance.

### 2.4. Endpoints and Data Analysis

#### 2.4.1. LC and Survival

Both LC and survival rate were calculated from the date of first re-irradiation fraction. Duration of LC and survival were analyzed by the actuarial method of Kaplan and Meier [22]. Local failure was defined as in-field relapse. Patients dying with LC, or alive with continuing LC at last follow-up (FU), were censored at the date of death or last follow-up, respectively.

Last FU was the last date with information on locoregional disease status. For overall survival patients known to be alive at last FU were censored at that date.

#### 2.4.2. Toxicity

Grade 3–5 late toxicities were assessed according to The National Cancer Institute’s Common Terminology Criteria for Adverse Events, (CTC–AE) version 3.0 [23]. To avoid bias, aggravation of pre-existing toxicity as well as toxicity of uncertain cause were considered to be related to the present treatment and scored accordingly. Toxicity was considered late when occurring >3 months after the start of reRT + HT. Late toxicity was calculated by the actuarial method of Kaplan and Meier [22] from the start of reRT + HT to the date of first ≥ grade 3 toxicity notification. Patients without late toxicity were censored at date of last FU. Six patients did not have data on late toxicity and were excluded from toxicity analysis but were included in all other analyses.

### 2.5. Statistics

Statistical analysis was carried out using the statistical program R version 2.13.0 and SPSS version 23 (SPSS Inc., Chicago, IL, USA). A multivariable analysis (Cox regression) was done for the LC and ≥ grade 3 late toxicity, using patient, disease, and treatment-related variables. All multivariable tests were carried out in backward Wald stepwise manner [24]. Only variables available for at least 80% of the population were included. The two-tailed Pearson correlation test was used to determine correlation coefficients. Variables with strong (>70%) correlations were not entered in the same multivariable model. The continuous variables were checked for linearity by using spline regression curves and spline coefficients tested for non-linearity. Variables included in the multivariable analysis for LC were: time interval from primary tumor to current recurrence, time interval from primary RT to reRT, time interval from current surgery to reRT, age, presence/history of distant metastases (DM), presence/history regional disease, tumor site, tumor size prior to surgery, lymphangitis, number of recurrence episodes, year of treatment, reRT regimen, current chemotherapy, and current endocrine treatment. Variables included in the analysis for ≥ grade 3 late toxicity: reRT field size, time interval from current surgery to reRT, age, type of current surgery, time interval from primary RT to reRT, previous/current chemotherapy, previous/current endocrine treatment, total number of different chemotherapy episodes, and year of treatment. The level of statistical significance was considered <0.05 for all analyses. 

## 3. Results

### 3.1. Treatment Compliance

Overall, the reRT + HT treatment was well tolerated. In Institute A, 97% of patients finished the treatment according to plan. Two patients did not finish treatment. One because of refusal and the second patient because of severe skin toxicity [18]. In Institute B the compliance rate was 99%; one patient refused the last reRT fraction. 

### 3.2. Clinical Outcome

For the group of patients from Institute A the reported 3- and 5-year infield local control (LC) rates were 78% and 65%, respectively, with a median FU time of 64 months [18] (Figure 2a). For the group of patients from Institute B the median overall FU time was 60 months. The current 3- and 5-year infield local control (LC) rates were and 76% and 70%, respectively (Figure 2b). Previously reported survival rates were 66% after 3 years and 49% after 5 years for the Institute A population ([18]] (Figure 2a), compared to 78% and 62% for the Institute B population in the current study (Figure 2b).

All variables mentioned in the materials and methods section were tested statistically for impact on their respective endpoints. Multivariable analyses were not conclusive for both study populations due to small patient numbers, though the same factors were indicated to be of influence in univariable analyses; time interval to recurrence and concurrent endocrine treatment were significant in both study populations (*p* = 0.001 and *p* = 0.04 for Institute A, *p* = 0.03 and *p* = 0.05 for Institute B, respectively). In multivariable analyses the only factor that remained a significant positive factor for LC in the Institute A population was a longer time interval to recurrence (*p* = 0.004, Hazard Ratio (HR) = 0.2) [18]. For the current population (Institute B) the only significant positive factor was concurrent endocrine treatment (*p* = 0.041, HR = 2.7).

### 3.3. Toxicity

The absolute ≥ grade 3 late toxicity rate was 43% in Institute A. The actuarial risk on ≥ grade 3 late toxicity after 3 years was 40% ([18]] (Figure 2a) [18]. In comparison, for the Institute B population the absolute rate was 18%, the actuarial risk was 15% after 3 years and 17% after 5 years (Figure 2b). Severe late toxicity consisted of mostly ulceration (17% of patients) in Institute A and of fibrosis (6% of patients) in Institute B.

In contrast to Institute B, 16 (21%) patients treated in Institute A experienced more than one severe complication. Three patients had ulcers progressing into bronchocutaneaous fistula. In one of these patients infections complicated the ulcer. In the other two patients ulceration was persistently progressive despite different interventions, without evidence of tumor presence [18]. No such cases were seen in Institute B.

Statistical analyses did not yield significant factors for the risk of severe late toxicity in either of the populations [18].

The number of specific ≥ grade 3 late toxicities for the Institute A and Institute B population are given in Table 2.

## 4. Discussion

Overall, the results of curative treatment for loco-regional recurrences remain disappointing. For surgery alone reported failure rates range from 4–37% for salvage mastectomy [25] to 48–76% [1,26] for local resection of chest wall recurrences. For reRT as sole treatment after surgery for locoregional recurrences, 1-year LC rates of 59–67% [27,28] have been reported, with a 3-year LC rate of 75% [29]. The number of patients and FU times in those studies were quite low and varied from 22 to 47 months. A larger study by Janssen (*n* = 83) [30] investigated the effect of reRT of 45 Gy (1.8 Gy per fraction) to the partial breast (*n* = 42) or mastectomy scar (*n* = 41) for patients with local recurrence either after the second breast-conserving therapy, or following mastectomy. After a median FU of 35 months the infield local relapse rate was 14.5%. Considering these data we believe our 5-year LC rate of 70% when combining reRT + HT after surgery is promising, taking into account the presence of risk factors for subsequent local relapse and resistance to previous treatments in our patient group.

When comparing our previously published study on Institute A patients [18] to the current analysis of Institute B patients we see many similarities regarding patient and tumor characteristics and duration of local control. Two points that are remarkably different are the survival rate (13% higher in Institute B, not significant) and the rate of ≥ grade 3 late toxicity (25% higher in the Institute A, *p* = 0.001). The difference in survival rate might be explained by a shorter median time interval to recurrence in Institute A compared to institute B (60 vs. 68 months). Though this difference between institutes was not significant, a shorter time interval had a significant negative effect on survival in both institutes and might be a sign of more aggressive disease (Institute A; *p* < 0.001, HR = 3, Institute B; *p* = 0.02, HR = 2.3).

Toxicity was more severe in the Institute A population (15% grade 4 complications, including two fistula, compared to 2% grade 4 complications in Institute B). This difference was also reported in our previous publications in which we analyzed both Institute A and Institute B patients. One of our studies reported results of 413 patients treated for irresectable recurrent breast cancer treated with reRT + HT. Although LC was not affected by the difference in reRT schedule and technique, the risk on overall ≥ grade 3 late toxicity was significantly increased (HR = 2, *p* = 0.023) with regimen A compared to regimen B with single or multiple electron fields. Five patients in the regimen A group died due to complications following ulceration [31]. Another study focused on the location of rib fractures when reRT + HT was given after surgery for recurrent breast cancer. All but one rib fracture occurred in the regimen A group and were located in the photon-electron abutments, with a maximum of eight fractures in one and the same patient [19]. As the treatments in both institutions came in fact as “packages”, each consisting of a number of elements of which several varied among the institutions it remains difficult to clearly identify the relative importance of each of the treatment-related factors on the occurrence of complications. Theoretically, this would best be examined in a prospective study. However, the Dutch protocol was carefully adapted in 2015, based on (among other aspects), the results of our analyses, and will not make this kind of toxicity acceptable in the future. The reRT schedule is now more fractionated (23 × 2 Gy to a total dose of 46 Gy) and patients are given Volumetric Modulated Arc Therapy (VMAT) with photons. Results must be awaited as the optimal schedule remains to be adjusted to individual sensitivity and toxicity from previous treatments, which should also be taken into account.

In recent years our HT population has shifted from mostly irresectable to 95% resectable patients. This shift also changes the whole clinical perspective. For this type of patient, endpoints such as quality of life (QoL), cosmetic outcome etc. become more and more important and need to be considered not only in today’s RT discussions, but with the inclusion of HT specialists. Figure 2b shows that at 5 years the risk of severe toxicity equals the chances of survival and eventually outruns both survival and local control chances with regimen A. The situation we wish for in the long term is much more likely to occur with regimen B (Figure 3).

Prospective studies and large-scale cooperation are necessary to optimize treatment for this population and reveal the individual impact of surgery, reRT, and hyperthermia and help to optimize the sequence of reRT + HT and, most importantly, whether they should be given before or after surgery. It might be worthwhile to investigate the effect of reRT + HT before surgery. This might be more effective as the studies have so far shown significant antitumor effect of HT added to RT on the gross tumor, but has never been tested for residual tumor.

Although the biological working mechanisms for hyperthermia are varied and more are elucidated each day [32], large randomized studies are falling behind. Recently a German prospective observational study was started with reRT + HT after R1/R2 resection for patients with recurrent breast cancer. reRT was given in 1.8–2 Gy fractions to a total dose of 50–50.4 Gy using tangential fields or one or multiple electron fields to the chest wall followed by an optional boost to 60–60.4 Gy [33] Each patient was scheduled to receive 11 HT sessions. This study, although potentially worthwhile, will not give us the answers we need to optimize reRT + HT treatment for our patients. It is again a small-sized non-randomized study from which ultimately no conclusions can be drawn, for instance about optimal reRT fraction size and/or of the effect of the individual treatment modalities and sequence hereof on treatment outcome and toxicity. Moreover, the same reRT schedule and techniques have already been used in the United States for patients with irresectable disease and showed low toxicity profiles [8].

In broader perspective, it seems there is some reluctance to offer reRT, which might be due to the lack of robust late toxicity data. To overcome this we need large-scale international cooperation to exchange and gather data, provide advice, and start-up large scale randomized trials focusing on the strengths of HT, which include increasing the effects of mainstream treatments to allow dose reductions and thereby decreased toxicity, without intrinsic toxicity of HT. An interesting option might be to combine reRT + HT with immunotherapy. It has already been demonstrated that immunotherapy when combined with RT + HT shows enhanced tumor responses [34]. Trimodaltity treatment might allow for reduction of reRT dose and hence less locoregional toxicity.

## 5. Conclusions

The combination of reirradiation and hyperthermia as consolidating treatment after macroscopically complete excision of loco-regional breast cancer recurrences is well tolerated and results in durable overall local control in patients at risk for locoregional recurrent breast cancer. Lowering the reRT fraction dose to ≤3 Gy and using limited-sized electron fields do not seem to compromise either LC or survival in this patient group compared to larger treatment volumes including elective irradiation within the wider margins. Large-scale international clinical studies are necessary for optimizing these kinds of combination treatments. 

## Figures and Tables

**Figure 1 cancers-11-00782-f001:**
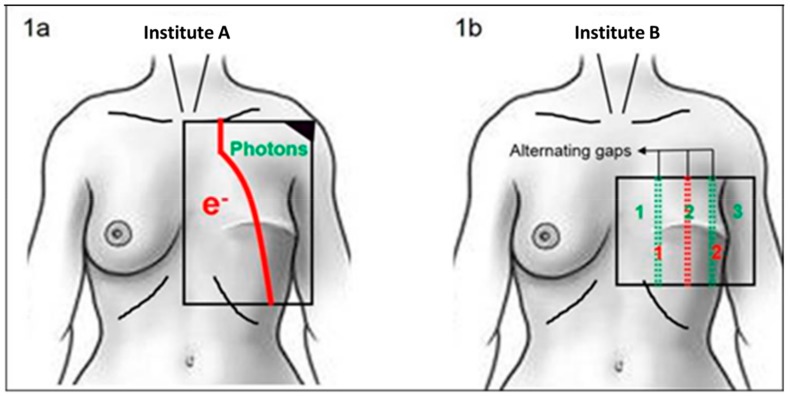
Standard chest wall radiation technique. (**a**) One anterior–posterior electron field abutted to one anterior–poster–posterior–anterior photon field. (**b**) Alternating use of two or three abutted anterior-posterior electron fields separated by either one or two small gaps.

**Figure 2 cancers-11-00782-f002:**
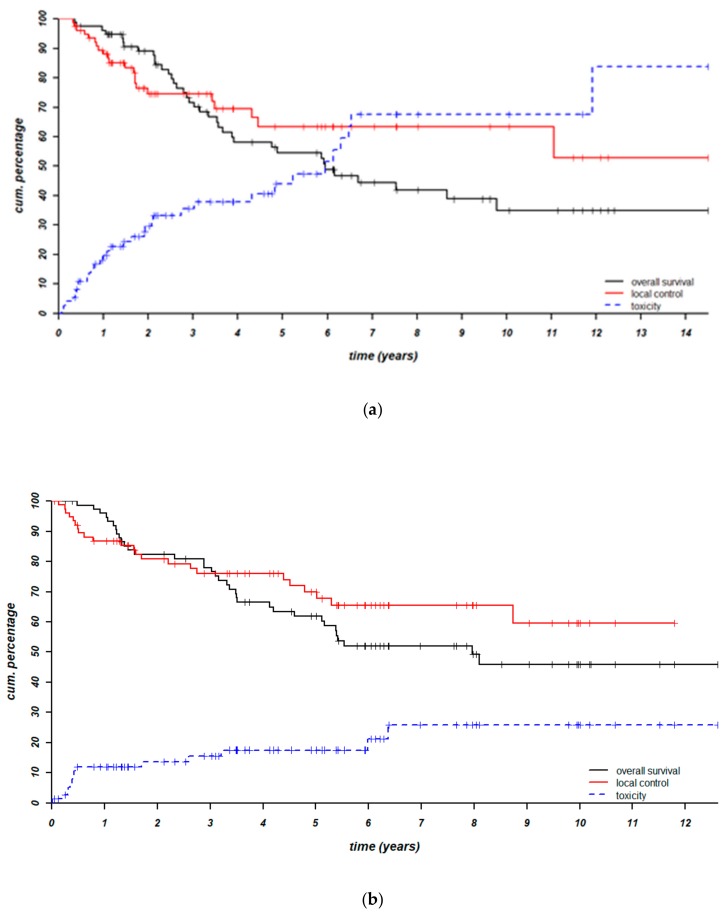
Local control, survival and ≥ grade 3 late toxicity risk after regimen A (**a**) and regimen B (**b**).

**Figure 3 cancers-11-00782-f003:**
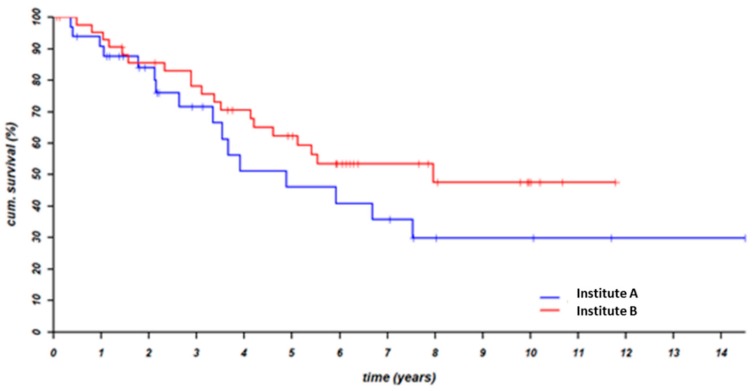
Survival without local infield progression nor ≥ grade 3 late toxicity.

**Table 1 cancers-11-00782-t001:** Patient and treatment characteristics.

Current Episode	Institute A [18]	Institute B
Number of patients	78	78
Median FU time	64 (8–151) months	60 (0.6–151) months
Median age at current treatment	52 (29–80) years	55 (35–78) years
Menopausal status ^a^		
Post	61 (78%)	52 (84%)
Median tumor size ^b^ (estimated)		
0–3 cm	40 (62%)	36 (56%)
>3 cm	25 (38%)	28 (44%)
Presence/history of distant metastases (DM)	6 (8%)	7 (9%)
Presence/history of regional disease	14 (18%)	8 (10%)
Presence/history of contralateral disease	12 (15%)	12 (15%)
Previous LR (1–6 episodes per patient)	34 (44%)	35 (45%)
Recurrence site		
Breast	34 (44%)	35 (45%)
Chest wall	37 (47%)	38 (49%)
Regional nodes	7 (9%)	5 (6%)
Median TI Surgery–ReRT	2 (0.9–8) months	2 (0.7–5) months
Median TI primaryRT–reRT ^c^	58 (10–241) months	66 (12–404) months
Median TI primary tumor-recurrence	60 (15–297) months	68 (8–403) months
Median primaryRT dose ^d^	50 (15–60) Gy	50 (45–50) Gy
Local or regional primaryRT boost^e^	63 (81%)	53 (69%)
Median boost dose	15 (5–25) Gy	16 (10–25) Gy
Surgery (R0/R1) ^f^	29 (39%)/45 (61%)	35 (53%)/31 (47%)
Salvage mastectomy	31 (40%)	35 (45%)
Chest wall resection (CWR)	6 (8%)	2 (3%)
Local excision	34 (44%)	36 (46%)
Other	7 (9%)	5 (6%)
ReRT dose ^g^		
12 × 3 (36) Gy	0	73 (94%)
8 × 4 (32) Gy	76 (97%)	1 (1%)
Other	2 (3%)	3 (4%)
ReRT technique		
Abutted photon + electrons field	60 (77%)	10 (13%)
Electron fields with gaps	0	20 (26%)
Single electron field	11 (14%)	41 (52%)
Other	7 (9%)	7 (9%)
Median electron-energy ^h^	8 (6–15) MeV^0^	9 (4–15) MeV
Median photon-energy ^i^	6 (5–14) MV^0^	6 (6–15) MV
Median reRT field size ^j^	4 (0.8–8.4) dm^2^	2 (0.7–7.6) dm^2^
Systemic treatment ^k^	39 (50%)	38 (49%)
Chemotherapy ^l^	14 (18%)	14 (18%)
Endocrine treatment ^m^	33 (43%)	31 (40%)

^a^ Institute B: missing for 16 patients. ^b^ Institute A: missing for 13 patients, Institute B: missing for 14 patients. ^c^ Institute B: missing for one patient. ^d^ Prior to reRT + HT; Institute B: missing for one patient. ^e^ Institute B: missing for one patient. ^f^ Institute A: missing for one patient, Institute B: missing or uncertain for 12 patients. ^g^ Institute B: missing for one patient. ^h^ Institute B: missing for one patient. ^I^ Institute B: missing for two patients. ^j^ Institute A: missing for seven patients, Institute B: missing for two patients. ^k^ In addition to the reRT + HT. ^l^ Institute A: missing for one patient. ^m^ Institute A: missing for two patients. ^0^ [19]. *Abbreviations:* FU = follow-up, LR = locoregional recurrence, TI = time interval; reRT = reirradiation; HT = hyperthermia.

**Table 2 cancers-11-00782-t002:** ≥ Grade 3 late toxicity events.

Toxicity	Institute A ^†^ [18] Grade 3/4	Institute B Grade 3/4
Ulceration	7/6	3/2
Fistula	1/2	
Blistering		1/0
Arm edema	5/1	
Frozen shoulder	7/0	
Fibrosis	7/0	5/0
Telangiectasia	8/0	3/0
Brachial plexopathy	7/0	
Osteonecrosis	6/1	
Osteoneuropathy	1/0	
Bone fractures	0/1	
Cardiomyopathy	1/0	

^†^ Data missing for three patients, 16 patients had more than one ≥ grade 3 late toxicity.

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
