# Peer review of "Impact of Technique and Schedule of Reirradiation Plus Hyperthermia on Outcome after Surgery for Patients with Recurrent Breast Cancer"

_cancers, 2019, doi:10.3390/cancers11060782_

Round 1
Reviewer 1 Report
The authors have studied a combinatorial approach of reirradiation and hyperthermia as consolidating treatment after macroscopically complete resection who were at high risk of cancer recurrences. Regarding the study design, recruiting/enrolling the patients and data analysis, I have following comments/questions that need to be addressed.
1, The demography and inclusion/exclusion criteria information are not adequately described in the Methods and Materials section. Did both sites have similar scale of inclusion criteria? The authors have not included adequate information for surgery results, when the tumors recurred, their pathological grades, stage of the disease, histology (tumor size, multi-centricity, diffuse tumor growth, angioinvasion or poor differentiation, immune-phenotype if studied. This results/information should be included.
2. Similarly, they could have provided more information for Endpoints of the study and severe or non-severe adverse events encountered in the study and how they were managed clinically. These issues should be elaborated in a separate paragraph.
3. Though multivariable analyses were not conclusive for data obtained from both sites because of small patient numbers, can the first six attributes listed on page 5- “Median T1 surgery-reRT through median boost dose” be analyzed as a separate group of attributes/or focused group and shown as correlates by plotting a separate K-M survival figure or describe the outcome in a separate graph? Please perform a comparative assessment for HR and P values of this group with the one already discussed in the manuscript.
Author Response
# reviewer 1
The authors have studied a combinatorial approach of reirradiation and hyperthermia as consolidating treatment after macroscopically complete resection who were at high risk of cancer recurrences. Regarding the study design, recruiting/enrolling the patients and data analysis, I have following comments/questions that need to be addressed.
1, The demography and inclusion/exclusion criteria information are not adequately described in the Methods and Materials section. Did both sites have similar scale of inclusion criteria? The authors have not included adequate information for surgery results, when the tumors recurred, their pathological grades, stage of the disease, histology (tumor size, multi-centricity, diffuse tumor growth, angioinvasion or poor differentiation, immune-phenotype if studied. This results/information should be included.
We thank the reviewer for this pertinent question. According to the Dutch National guidelines, all patients with locoregional breast cancer recurrences in previously irradiated areas should be treated with reirradiation + hyperthermia, preferably preceded by surgery. The inclusion criteria of both institutes, consequently, are the same. We included all these patients retrospectively, without exclusion. We added this information in the text of the manuscript.
We only reported disease characteristics known for at least 80% of the population to be able to explore correlations with the endpoints. Unfortunately, despite our repetitive follow-up requests to referring specialist, only a minority of the details about the performed surgery was recovered. Furthermore, some patients were treated before the time that some of today’s pathology analyses like HER2-neu receptor status were routinely performed. Finally, almost 50% of our patients had several surgery interventions; either for their second, third, fourth or even fifth recurrence, hence pathology was only used to confirm presence of disease or not. We now reported this information more extensively in the text of the manuscript
2. Similarly, they could have provided more information for Endpoints of the study and severe or non-severe adverse events encountered in the study and how they were managed clinically. These issues should be elaborated in a separate paragraph.
We agree with this constructive suggestion, Unfortunately, due to the retrospective character of our study, this data was not reported or could not be retrieved from the available paper files. We focused on severe late toxicity as main endpoint because of it’s major impact on patients quality of live, in comparison to mild toxicity. Therefore, we have put our emphasis on the burden of the treatment in respect to the beneficial effects (LC/survival). We added this to the text of the manuscript.
3. Though multivariable analyses were not conclusive for data obtained from both sites because of small patient numbers, can the first six attributes listed on page 5- “Median T1 surgery-reRT through median boost dose” be analyzed as a separate group of attributes/or focused group and shown as correlates by plotting a separate K-M survival figure or describe the outcome in a separate graph? Please perform a comparative assessment for HR and P values of this group with the one already discussed in the manuscript.
Median TI surgery – reRT | 2 (0.9-8) months | 2 (0.7-5) months |
Median TI primaryRT – reRTc | 58 (10-241) months | 66 (12-404) months |
Median TI primary tumor- recurrence | 60 (15-297) months | 68 (8-403) months |
Median primaryRT dosed | 50 (15-60) Gy | 50 (45-50) Gy |
Local or regional primaryRT booste | 63 (81%) | 53 (69%) |
Median boost dose | 15 (5-25) Gy | 16 (10-25) Gy |
All time interval variables correlate and as such cannot be put in one and the same multivariable model. On univariable analysis, time interval to recurrence and time interval to reRT (high correlation) are significant for LC and survival in both institutes, whereas the time interval between surgery and reirradiation has no significant effect on either endpoints. The three other variables relate to the primary RT, if tested they fail to result in a significant effect on any of the endpoints. We did not include this in the manuscript.
Reviewer 2 Report
The manuscript wrote by Oldenborg et al. describes the use of reirradiation with hyperthermia as a therapeutic treatment in addition to the patient’s surgery. Local control, survival rate, and toxicity result and clinical observation are demonstrated. The reRT+HT could be an effective treatment plan for a patient is also described.
The manuscripts present a solid amount of clinical work, largely well elaborated. At the same time, in some aspects, the manuscript coherence and the significance of demonstrated results are lacking, in my opinion, for publication in Cancers after revision.
Three main points to illustrate the statement above:
1) How to explain the result taken all the variations into considering is not quite clear. What is the importance of the chest wall radiation technique, RT treatment frequency, total dose, adjusted energy, reRT field size, the time interval from current surgery to reRT. Authors are recommended to look into other possible variations more carefully by providing a favorable comparison between the Institute A and institute B groups.
2) Is the high toxicity in institute A expected? More insightful research should have been done on the reRT+HT technique. That discussion should be added to the introduction part of the manuscript to support the scientific rationale of the study.
3) What are the proposed solutions to reduce toxicity? The use of immunotherapy might elevate the therapeutic effect of RT+HT and reduce the side effect. Reference about immunotherapy and hyperthermia can be referred, Cancers 10.12 (2018): 469; Science immunology 2 (17), eaan5692-eaan5692; Int J Hyperthermia. 2014 Dec;30(8):531-9.
Author Response
# reviewer 2
The manuscript wrote by Oldenborg et al. describes the use of reirradiation with hyperthermia as a therapeutic treatment in addition to the patient’s surgery. Local control, survival rate, and toxicity result and clinical observation are demonstrated. The reRT+HT could be an effective treatment plan for a patient is also described.
The manuscripts present a solid amount of clinical work, largely well elaborated. At the same time, in some aspects, the manuscript coherence and the significance of demonstrated results are lacking, in my opinion, for publication in Cancers after revision.
Three main points to illustrate the statement above:
1) How to explain the result taken all the variations into considering is not quite clear. What is the importance of the chest wall radiation technique, RT treatment frequency, total dose, adjusted energy, reRT field size, the time interval from current surgery to reRT. Authors are recommended to look into other possible variations more carefully by providing a favorable comparison between the Institute A and institute B groups.
Thank you for your constructive appraisal of our work. In fact, we agree with this reviewer’s remark: the treatments in both institutions came in fact as “packages”, each consisting of a number of elements of which several varied among the institutions. As important factors that could be separately evaluated, we recall the effects of abutting fields and fraction size that has been addressed for rib fractures in one of our previous papers (Rib fractures after reirradiation plus hyperthermia for recurrent breast cancer; predictive factors. Strahlenter Onkol (2016) 192: 240-274). Overall however, it remains difficult to clearly identify the relative importance of each of the treatment-related factors on the occurrence of complications. Theoretically, this would best be examined in a prospective study but the current (Dutch) protocol was carefully adapted based on among others the results of our analyses and won’t make this type of treatment acceptable anymore. We have now stated this clearly in the discussion.
2) Is the high toxicity in institute A expected? More insightful research should have been done on the reRT+HT technique. That discussion should be added to the introduction part of the manuscript to support the scientific rationale of the study.
We agree with the reviewer, however prospective clinical studies on hyperthermia and reRT were and are difficult, due to the small number of hyperthermia institutes in the world and the limited availability of commercial equipment. On top of this, it would be quite impossible to obtain sufficient patient accrual to reliably address this issue. Furthermore, back in the early days of hyperthermia around 1980, RT and HT techniques were less sophisticated and knowledge as well as experience far more limited. Initially even more than now especially patients who had no other options left had to be treated. They were at higher risk to develop severe side effects following reRT+HT as a cumulative result of the reRT+HT after previous treatments, probably also combined with individual sensitivity to toxicity. However, the addition of HT to any treatment has proven not to result in a significant increase of toxicity. We made a statement about all this in the introduction.
3) What are the proposed solutions to reduce toxicity? The use of immunotherapy might elevate the therapeutic effect of RT+HT and reduce the side effect. Reference about immunotherapy and hyperthermia can be referred, Cancers 10.12 (2018): 469; Science immunology 2 (17), eaan5692-eaan5692; Int J Hyperthermia. 2014 Dec;30(8):531-9.
We agree with the reviewer that this would be a very interesting and promising topic for future clinical research. Solutions to reduce toxicity from reRT are stated in the discussion and now include the option of combinations with immunotherapy including the reference from 2018.
Reviewer 3 Report
Interesting paper
Orignal paper. Methods and results are clear, Conclusion logic and complete.
Minor
Authors should describe more carefully the limitations of their study based on the "coincidental" back to back comparison.of 2 different treatement regimens in 2 institutions in a similar time period with the same number of patients.
Written consent should be obtained from medical leadership of both institutions for this comparison and the content of this manuscript
Author Response
# reviewer 3
Interesting paper
Orignal paper. Methods and results are clear, Conclusion logic and complete.
Minor
Authors should describe more carefully the limitations of their study based on the "coincidental" back to back comparison.of 2 different treatement regimens in 2 institutions in a similar time period with the same number of patients.
We thank reviewer 3 for this very positive appraisal. For the suggestion, with which we agree, we would like to refer to our reply to reviewer 1.
Written consent should be obtained from medical leadership of both institutions for this comparison and the content of this manuscript.
We think we have covered this issue by sending, by email, a complete copy of the manuscript to the former and present chairmen of the radiation oncology / hyperthermia departments of both institutes two weeks prior to submission. They were informed on submission date and asked for comments or suggestions., if they had any. All suggestions were then added to the manuscript. None of them disagreed with submission of the content.